# Phase Conductance of BiFeO_3_ Film

**DOI:** 10.3390/s23229123

**Published:** 2023-11-11

**Authors:** Yufeng Wang, Peng Zhou, Leonid Fetisov, Yuri Fetisov, Yajun Qi, Tianjin Zhang

**Affiliations:** 1Ministry of Education Key Laboratory for Green Preparation and Application of Functional Materials, Hubei Provincial Key Laboratory of Polymers, Collaborative Innovation Center for Advanced Organic Chemical Materials Co-Constructed by the Province and Ministry, School of Materials Science and Engineering, Hubei University, Wuhan 430062, China; wang_yf_99@163.com (Y.W.); zhangtj@hubu.edu.cn (T.Z.); 2Research-Education Center “Magnetoelectric Materials and Devices”, MIREA—Russian Technological University, Moscow 119454, Russia

**Keywords:** local conductance, tetragonal-like phase, rhombohedral-like phase, electric poling

## Abstract

In this work, the local conductance of the tetragonal-like (T-like) and rhombohedral-like (R-like) phases of epitaxial BiFeO_3_ film is systematically studied via conductive atomic force microscopy. At higher tip voltage, there is a mutual transition between the T-like and R-like phases, which could be attributed to the strain relaxation in the T-like phase induced by electric poling, as well as local polarization switching. The T-like phase exhibits a higher conductance, which is related to the lower interface potential barrier between the tip and film surface. Reversible low- and high-current states in the T-like phase can be tuned by polarization switching. These results will be helpful for designing novel nanoelectronic devices, such as voltage and strain sensors.

## 1. Introduction

Bismuth ferrite (BiFeO_3_, denoted as BFO) is one of the very few room-temperature single-phase multiferroic materials that have potential applications in the fields of nonvolatile memories, spintronic, and ferroelectric devices [1,2]. The investigations related to BFO include, but are not limited to, the finding of self-regulated chemical substitution in a highly strained BFO [3], the demonstration of coexisting morphotropic phase boundary and giant strain gradient in BFO films [4], and the revealing of flexoelectricity in the mixed-phase regions of epitaxial BFO thin films [5]. Electrical properties in the microscopic area of BFO films have been intensively investigated in recent years [6,7,8]. For instance, the enhanced injection currents induced by polarization switching in BFO thin films were observed via conductive atomic force microscopy (c-AFM), where the injected current could be effectively modulated by applying a mechanical force. This mechanical modulation of the injected current was attributed to the mechanical force-induced changes in the barrier height and interfacial layer width [9]. The charged domain walls in BFO show a number of new functionalities due to their enhanced local conductivity. The link between the local structural distortions and charge screening phenomena in 109° tail-to-tail domain walls of BFO was elucidated [10]. More precisely, the intrinsic conductivity of 71° and 109° domain walls was revealed by probing the local conductance in BFO films, where the 71° domain wall exhibited an inherently charged nature, whereas the 109° domain wall was close to neutral. The different conductivities of the two kinds of domains were related to the bound-charge-induced bandgap reduction [11]. By manipulating the orientation of the ferroelectric domain walls, the interface in the BFO/TbScO_3_ heterostructure was conducted in the direction parallel to the domain stripes, but was insulated in the direction perpendicular to the domain stripes [12].

Similar to BFO, the investigations on domain switching in microscopic areas were also conducted for other materials. The impact of electrical and mechanical manipulations on the domains in (111)-oriented PbZr_0.2_Ti_0.8_O_3_ thin films with nano-twinned ferroelectric domain structures was examined [13]. A picture of the inter-relationship between piezoelectricity, flexoelectricity, mechanical manipulation, and domain switching in PbZr_0.2_Ti_0.8_O_3_ thin film was developed. In the epitaxial heterostructure of BaTiO_3_/SrRuO_3_, domain switching induced by a tip force of 320 nN was reported. This low mechanical threshold was related to the small compressive strain, the low oxygen vacancy concentration in the BaTiO_3_ film, and the high conductivity of the SrRuO_3_ electrode [14]. Recently, ferroelectric domains in PMN-PT crystals were found to be manipulated using an optical method [15]. Until now, studies on phase conductivity in BFO films, especially for systematic studies on the evolution of phase conductivity with tip voltage, are still lacking, though we can find various theoretical and experimental works in terms of domain wall conductivity in BFO [16,17,18].

In this work, the conductance of tetragonal-like (T-like) and rhombohedral-like (R-like) phases in epitaxial BFO films was investigated via c-AFM. The evolution of the topography and current of T-like and R-like phases with tip voltage were studied systematically. The mechanisms of the different electrical properties of T-like and R-like phases, as well as the poling dependence of current for the T-like phase, were also discussed. This work provides a clear picture of the phase conductance of BFO films under an electric field.

## 2. Materials and Methods

Epitaxial BFO film with a thickness of 70 nm was grown on La_0.7_Sr_0.3_MnO_3_ (LSMO)-buffered (001) LaAlO_3_ (LAO) single-crystal substrate via pulsed-laser deposition with a KrF excimer laser (λ = 248 nm). Details of the deposition procedure as well as the structural analyses are described elsewhere [19,20]. The electrical properties of BFO films were measured using c-AFM and piezoelectric force microscopy (PFM) (Asylum Research, MFP-3D Origin) at room temperature.

## 3. Results

Figure 1a shows the schematic diagram of the sample structure. During the c-AFM measurement, a negative voltage was applied on the film surface through the tip, while LSMO was connected to a positive voltage. The topography (Figure 1c) of the BFO film does not show any particles or contamination, demonstrating that the film is of high quality with a flat surface. The striped features observed in Figure 1c reveal different heights for the yellow and black areas. The transmission electron microscopy (TEM) results in our previous work confirmed that the BFO film was epitaxially grown on an LSMO-buffered LAO substrate [19,20]. T-like and R-like phases were also observed via TEM. Moreover, the two phases exhibit different lattice parameters and piezoelectric responses [21]. Therefore, the striped features in Figure 1c are related to the mixed phases in the BFO film, i.e., the T-like phase and R-like phases. The surface polarization directions along the out-of-plane and in-plane can be distinguished based on Figure 1d,e, respectively. Ferroelectric polarization switching for both T-like and R-like phases was proved via PFM amplitude and phase, as illustrated in Figure 1b (only the results of the T-like phase are shown). The T-like and R-like phases exhibited similar ferroelectric properties, with a coercive field of around 8 V (only the results of the T-like phase are shown in Figure 1b).

To investigate the evolution of the phase fraction and conductance of T-like and R-like phases under various tip voltages, c-AFM was used to measure both the topography and current of the BFO film. The areal fraction of the T-like phase for 70 nm BFO film was estimated to be 35% via transmission electron microscopy in our previous work [19,20]. Gray stripes in the topographic images denote R-like phases, whereas the remaining areas represent T-like phases. When the tip voltage is lower than 4 V, there is no obvious current difference between the T-like and R-like phases, as shown in Figure 2. However, the areal fraction of the R-like phase increases with an increase in the tip voltage, as indicated by the red and green circles in Figure 2a,b, respectively. The BFO film needs sufficient strain to stabilize its T-like phase [19,20], and a higher tip voltage can reduce the strain in the T-like phase via electric poling, resulting in relaxation of the T-like phase, as well as the transition from the T-like phase to R-like phase. When the tip voltage reaches 5 V, topography (Figure 2c) almost remains the same as the one with a tip voltage of 4.7 V, whereas the current map in Figure 2g exhibits a contrast. The current in the blue circles in Figure 2g is much higher than the one in the remaining area. By increasing the tip voltage to 5.1 V, a change simultaneously occurs in the topography and current of the BFO film, as shown in Figure 2d,h. From the comparison between Figure 2c,d, we can observe a transition from the R-like phase to the T-like phase, as the R-like phase cracks and disappears after applying 5.1 V. Polarization switching could be responsible for the shift from the R-like phase to the T-like phase. On the other hand, the areal fraction with a higher current in Figure 2h is larger than that in Figure 2g. The difference in current between the T-like and R-like phases can be observed in the orange circle in Figure 2h, revealing that the T-like phase has a higher current than the R-like phase.

When the tip voltage is decreased to 0, the areal fractions of T-like and R-like phases remain constant. The current of both phases does not show any difference in this case, as shown in Figure 3a,e. As the tip voltages are lowered to −4 V and −5 V, there is a mutual transition between T-like and R-like phases, indicated by boxes with different colors in Figure 3b,c. From the contrast within the red box and outside the green box, as well as the area within the orange box, there is a transition from an R-like phase to a T-like phase when the tip voltage is decreased from −4 V to −5 V. In contrast, we see a transition from a T-like phase to an R-like phase in the area within the red box. In Figure 3g, the dark areas indicate that the R-like phase has a lower current, which is consistent with the result shown in Figure 2h. As the tip voltage returns to 3.6 V, the current contrast between the T-like and R-like phases is more obvious, as shown in Figure 3h. The areal fractions of the topography for both phases are almost the same as those in Figure 3c.

The evolution of the T-like and R-like phases of the BFO film at different tip voltages is summarized in Figure 4. The gray and yellow rectangles represent the R-like and T-like phases, respectively. At a lower tip voltage, the T-like phase is inclined to shift to the R-like phase (below 4.7 V). At a relatively higher tip voltage, there is a mutual transition between the two phases. The T-like phase shows a higher conductivity than that of the R-like phase. Similar electronic conductivities of ferroelectric domain walls in self-assembled BFO nano-islands were investigated, where stable and repeatable on-and-off switching of conductive domain walls within topologically confined vertex domains was controlled by an electric field [22]. This on-off switching is attributed to the reversible transformations between the charged and neutral domain walls. Another method based on the scanning-probe technique was proposed to directly detect the specific conductivity of 71° ferroelectric domain walls in BFO, where the 71° walls showed a specific conductivity of 0.05 S/m [8]. By using atomic-scale imaging and in situ manipulation techniques, as well as high-resolution scanning transmission electron microscopy, the effect of oxygen vacancies on the domain wall conductivity can be understood comprehensively [23,24].

To further investigate the electrical properties of the T-like and R-like phases of BFO films, we rescanned the film surface using c-AFM with a tip voltage of 3.6 V, as shown in Figure 5a,b. The topography resembles that shown in Figure 3d, although there is little change in the current map. The outline of the current along the dotted line in Figure 5b is illustrated in Figure 5c. This dotted line covers the stripes of the T-like and R-like phases, indicating that different phases exhibit distinct conductivity. Figure 5d shows I–V loops of the T-like and R-like phases collected from the spots in Figure 5b, as indicated by the red arrows. Obviously, the current in the T-like phase is higher than that in the R-like phase. It was reported that there are two kinds of conduction mechanisms in ferroelectric materials, i.e., bulk effect and surface effect [25,26]. To gain insight into the conduction behavior of the BFO film, the possible mechanisms, such as Fowler–Nordheim tunneling, Schottky emission, Richardson–Schottky–Simmons emission, and Poole–Frenkel emission, were utilized to fit the I–V loops in Figure 5d. The experimental data can be fitted well with the Richardson–Schottky–Simmons emission model, the formula for which is as follows [26]:$$\ln\left(\frac{I}{T^{3/2}V}\right)∼-\frac{q\varphi_{B}}{kT}+\frac{\sqrt{q^{3}/4\pi \varepsilon_{0}\varepsilon_{opt}d}}{kT}V^{1/2}$$
where *φ_B_* and *ε_opt_* are the interface potential barrier and optical dielectric permittivity, respectively. The fitted values of *φ_B_* and *ε_opt_* for the R-like phase are 0.99 eV and 5.735, respectively, and for the T-like phase, the values are 0.961 eV and 2.872, respectively. Therefore, the higher conductivity in the T-like phase could be attributed to the lower interface potential barrier between the film surface and tip.

To demonstrate the correlation between polarization switching and the conduction of the T-like phase, writing pulses of +/−8 V were applied to the spot (Figure 5b) of the T-like phase for 1 s. After that, the I–V loops were measured, as shown in Figure 6a. For +8 V poling, the I–V curves are nearly the same when the tip scans from 0 to +6 V and from +6 V to 0. For −8 V poling, however, a sudden increase of current occurs at a voltage of around 5.5 V, during the increase of tip voltage from 0 to +6 V. When the tip scans from +6 V to 0, the I–V curve is similar to the that obtained +8 V poling. On the other hand, a reading bias of 4 V was applied to measure the current values after each writing pulse, as illustrated in Figure 6b. The T-like phase stays in a low current state after −8 V poling, whereas it stays in a high current state after +8 V poling. This reversible switching between low- and high-current states demonstrates switching between two conducting states with an electric field in the T-like phase of BFO film, which provides direct evidence for the relationship between polarization switching and local conductance. Similar results have been reported previously [25].

## 4. Discussion

The purpose of this study is to investigate the local conductance of BFO films. T-like and R-like phases are formed due to the different strains induced in perovskite BFO. Specifically, BFO is inclined to form an R-like phase at tensile to moderate compressive strains, whereas it forms a T-like phase when the compressive strain exceeds ~−4.5% [27]. It is intriguing to study the phase conductance when both the T-like and R-like phases exist in a single film. The strain varies when an electric field is applied on a local area of a BFO film, as the electric field changes the polarization and strain. With different magnitudes of positive or negative tip voltages applied on the film surface, the T-like and R-like phases should exhibit distinct responses. This is why the T-like phase shows a higher current when the tip voltage reaches specific values. Moreover, a phase transition occurs because of the strain changes. We observed low- and high-current states after applying negative and positive tip voltages, as the interface potential barrier is dependent on polarization. The shift between two different conducting states tuned by polarization can be used in nanoelectronic devices, such as memories.

## 5. Conclusions

In summary, the local conductance of the T-like and R-like phases of epitaxial BFO films has been systematically studied via c-AFM. At higher tip voltage, there is a mutual transition between the T-like and R-like phases, which could be attributed to the strain relaxation in the T-like phase induced by electric poling, as well as local polarization switching. The T-like phase exhibits a higher conductance, which is related to the lower interface potential barrier between the tip and film surface. Reversible low- and high-current states in the T-like phase tuning via polarization switching have been demonstrated.

As the conductance of the T-like and R-like phases is sensitive to voltage, this BFO film is a promising material for the design of voltage sensors. On the other hand, polarization switching can be realized via strain, and there is a relationship between phase conductance and strain. The BFO film can also be used in the field of strain sensors.

## Figures and Tables

**Figure 1 sensors-23-09123-f001:**
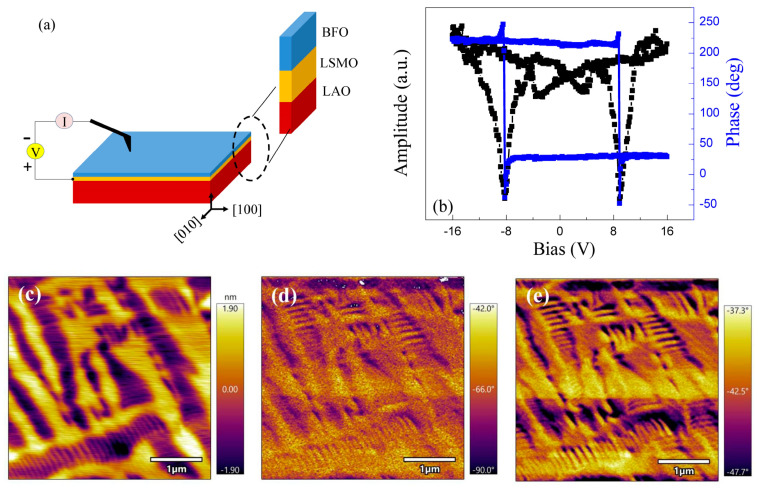
(**a**) Schematic diagram of the sample structure; negative voltage was applied on the film surface through the tip, while LSMO was connected to a positive voltage. (**b**) PFM amplitude (black) and phase (blue) of T-like phase of the BFO film. (**c**) Topography of the BFO film, (**d**) in-plane PFM phase, (**e**) and out-of-plane PFM phase of the BFO film.

**Figure 2 sensors-23-09123-f002:**
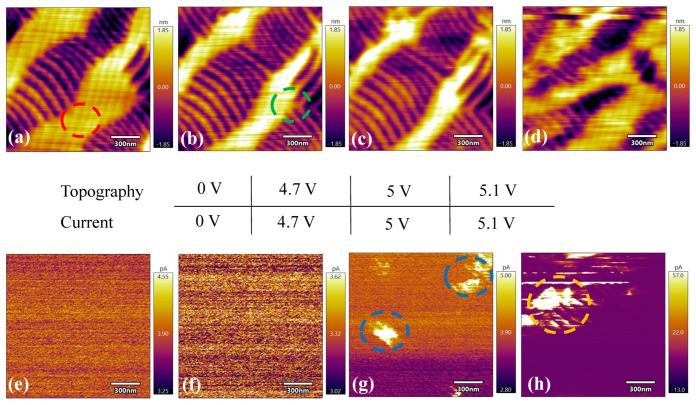
Topographies (**top panel**) and current maps (**bottom panel**) for tip voltages of 0 V (**a**,**e**), 4.7 V (**b**,**f**), 5 V (**c**,**g**), and 5.1 V (**d**,**h**). Scanning was performed on the same area at different tip voltages.

**Figure 3 sensors-23-09123-f003:**
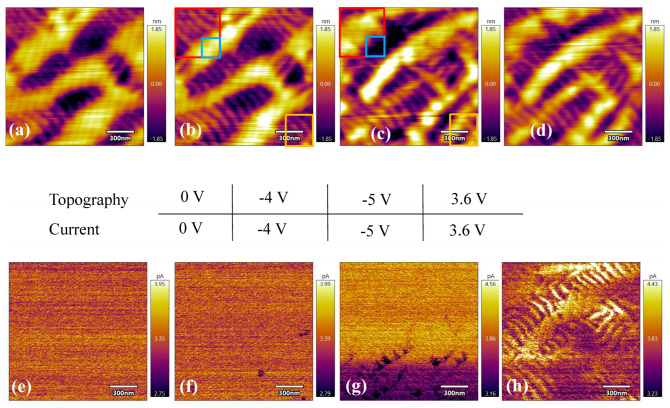
Topographies (**top panel**) and current maps (**bottom panel**) for tip voltages of 0 V (**a**,**e**), −4 V (**b**,**f**), −5 V (**c**,**g**), and 3.6 V (**d**,**h**). Scanning was performed on the same area at different tip voltages.

**Figure 4 sensors-23-09123-f004:**
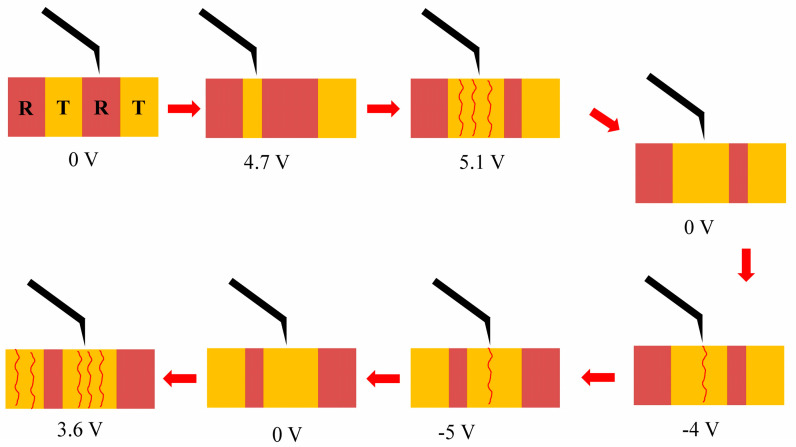
Summary of the evolution of phase transition and conductance under various tip voltages for BFO films, where T-like and R-like phases are denoted in different colors. Current is denoted by the wave lines.

**Figure 5 sensors-23-09123-f005:**
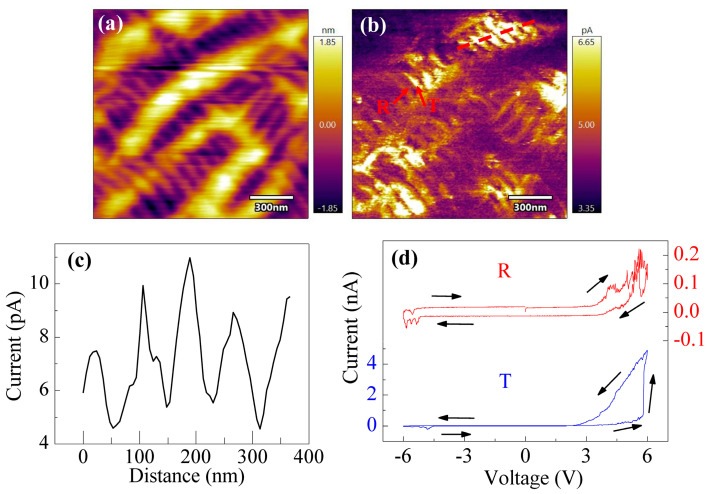
(**a**) Topography and (**b**) current map of the BFO film scanned with a tip voltage of 3.6 V. (**c**) Current outline collected along the red dotted line in (**b**). (**d**) Current–voltage (I–V) loops of T-like and R-like phases measured from the spots that are indicated by red arrows.

**Figure 6 sensors-23-09123-f006:**
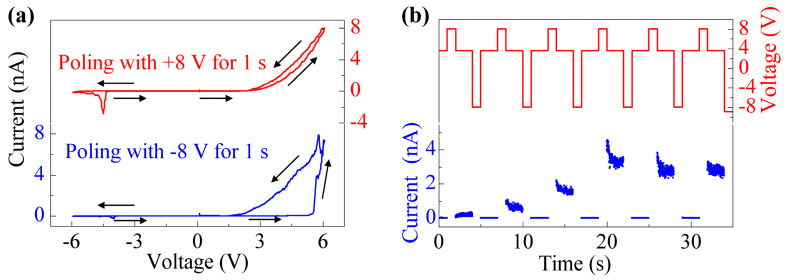
(**a**) I–V loops with poling voltages of +/−8 V. (**b**) Time dependence of current at different writing voltages. Date was collected in the T-like phase.

## Data Availability

Data are contained within the article.

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
