# Peer review of "Phase Conductance of BiFeO3 Film"

_sensors, 2023, doi:10.3390/s23229123_

Round 1
Reviewer 1 Report
Comments and Suggestions for Authors
In the manuscript “Phase conductance of BiFeO3 film”, BiFeO3 thin film is characterized by piezoelectric force microscopy (PFM) and conductive atomic force microscopy (C-AFM). Two phases were observed in epitaxial BFO film: tetragonal-like (T-like) and rhombohedral-like (R-like), which change according to the tip voltage applied. The main conclusion of this paper is that because the interface potential barrier is dependent on polarization, low and high current states are observed after applying opposite tip voltages.
In my opinion, the manuscript can be published in Sensors after minor corrections.
- The caption in Figure 1 must be modified to better describe it; even the text referring to the figure is not clear. Figures 2, 3, and 4 also have too-short captions.
- The authors must explain in the text why the stripes observed in Fig. 1(c) reveal the mixed phase in BFO film, also providing a reference. This is fundamental to understanding the findings in Fig. 3 and the model in Fig. 4.
Comments on the Quality of English Language
The authors must improve the English of the text, which contains minor errors.
Author Response
Reviewer 1
In the manuscript “Phase conductance of BiFeO3 film”, BiFeO3 thin film is characterized by piezoelectric force microscopy (PFM) and conductive atomic force microscopy (C-AFM). Two phases were observed in epitaxial BFO film: tetragonal-like (T-like) and rhombohedral-like (R-like), which change according to the tip voltage applied. The main conclusion of this paper is that because the interface potential barrier is dependent on polarization, low and high current states are observed after applying opposite tip voltages.
In my opinion, the manuscript can be published in Sensors after minor corrections.
Answer: We would like to thank the reviewer for his/her carefully reading the manuscript and providing positive comments. We thank the reviewer for his/her sincere recommendations. We believe the comments proposed by the reviewer are very helpful to improve our manuscript. The issues have been addressed and answered one by one as follows:
- The caption in Figure 1 must be modified to better describe it; even the text referring to the figure is not clear. Figures 2, 3, and 4 also have too-short captions.
Answer: Thank you for your valuable comments and suggestions. The captions for figures 1-4 have been revised. The description for figure 1 in the main text has been revised also. Please see the manuscript, where the revised contents were marked in red.
- The authors must explain in the text why the stripes observed in Fig. 1(c) reveal the mixed phase in BFO film, also providing a reference. This is fundamental to understanding the findings in Fig. 3 and the model in Fig. 4.
Answer: Thank you for your comments. We have discussed the reasons on mixed phases in BFO film as follows and in the main text. The related references have been provided.
The striped features observed in figure 1(c) reveal different height for yellow and black areas. Transmission electron microscopy (TEM) results in our previous work confirmed BFO film was epitaxially grown on LSMO-buffered LAO substrate [Applied Physics Letters 99 (13): 132905 (2011); Applied Surface Science 425, 117-120 (2017)]. T-like and R-like phases were also observed via TEM. Moreover, the two phases exhibit different lattices parameters and piezoelectric response [Applied Physics Letters 97 (24): 242903 (2010); Applied Physics Letters 96 (25): 252903 (2010)]. Therefore, the striped features in figure 1(c) are related to the mixed phases in BFO film, i.e., T-like phase and R-like phase.
- The authors must improve the English of the text, which contains minor errors.
Answer: Thank you for your comment and suggestion. We have proofread the English in the main text.

Reviewer 2 Report
Comments and Suggestions for Authors
The authors present a study on phase conductance of BiFeO3 film with R- and T-like phases. The study is interesting, and i only have a few comments as listed below:
The introduction should include relevant literature on other systems showing different behavior depending on crystal structure.
The final paragraph of the introduction (lines 47-56) should not present the results, but instead it should present what will be investigated in this paper.
The BFO structure and purity should somehow be evidenced, e.g., by XRD.
In the discussion section, the authors should include more studies to compare against their own observations.
Author Response
Reviewer 2
The authors present a study on phase conductance of BiFeO3 film with R- and T-like phases. The study is interesting, and I only have a few comments as listed below:
Answer: We would like to thank the reviewer for his/her carefully reading the manuscript and providing positive comments. We thank the reviewer for his/her sincere recommendations. We believe the comments proposed by the reviewer are very helpful to improve our manuscript. The issues have been addressed and answered one by one as follows:
- The introduction should include relevant literature on other systems showing different behavior depending on crystal structure.
Answer: We have included relevant literature on other systems in the main text (and as follows). The revised contents were marked in red.
Similar to BFO, the investigations on domain switching in microscopic area were also conducted in other materials. The impact of electrical and mechanical manipulations on domains in (111)-oriented PbZr0.2Ti0.8O3 thin films with nano-twinned ferroelectric domain structures was examined [Advanced Functional Materials 31 (19): 2011029 (2021)]. A picture of the interrelationship between piezoelectricity, flexoelectricity, mechanical manipulation, and domain switching in PbZr0.2Ti0.8O3 thin film was developed. In epitaxial heterostructure of BaTiO3/SrRuO3, domain switching induced by tip force of 320 nN was reported. This low mechanical threshold was related to the small compressive strain, the low oxygen vacancy concentration in BaTiO3 film, and the high conductivity of SrRuO3 electrode [ACS Applied Materials & Interfaces 14 (43): 48917-48925 (2022)]. Recently, ferroelectric domains in PMN-PT crystal were found to be manipulated by an optical method [Advanced Optical Materials 10 (21): 2201092 (2022)].
- The final paragraph of the introduction (lines 47-56) should not present the results, but instead it should present what will be investigated in this paper.
Answer: Thank you for your comment and suggestion. We have revised the final paragraph of the introduction, as shown below and in the main text.
In this work, conductance of tetragonal-like (T-like) and rhombohedral-like (R-like) phases in epitaxial BFO film was investigated via c-AFM. Evolution of topography and current of T-like and R-like phases with tip voltage were studied systematically. The mechanisms on different electrical properties for T-like and R-like phases, as well as poling dependence of current for T-like phase, were also discussed. This work provides a clear picture of phase conductance of BFO film with electric field.
- The BFO structure and purity should somehow be evidenced, e.g., by XRD.
Answer: Thank you for your comment and suggestion. Actually, we have conducted transmission electron microscopy (TEM) for BFO/LSMO/LAO heterostructure used in this work, the related results have been reported in our previous work [Applied Physics Letters 99 (13): 132905 (2011); Applied Surface Science 425, 117-120 (2017)], which confirms the epitaxial growth of BFO film, as well as the T-like and R-like phases in BFO film. Therefore, we didn’t show any results of X-ray diffraction or TEM in the main text. We focused on conductance of T-like and R-like phases in BFO film.
- In the discussion section, the authors should include more studies to compare against their own observations
Answer: Thank you for your comment and suggestion. We have included some studies in the main text to compare against our observations, as shown below.
Similar electronic conductivities of ferroelectric domain walls in self-assembled BFO nano-islands were investigated, where stable and repeatable on-and-off switching of conductive domain walls within topologically confined vertex domains was controlled by electric field [Nat Commun 13 (1): 3255 (2022)]. This on-off switching is attributed to the reversible transformations between charged and neutral domain walls. Another method based on scanning-probe technique was proposed to directly detect the specific conductivity of 71° ferroelectric domain walls in BFO, where the 71° walls showed a specific conductivity of 0.05 S/m [ACS Applied Electronic Materials 4 (6): 2739-2746 (2022)]. By using atomic-scale imaging and in situ manipulation techniques, as well as high-resolution scanning transmission electron microscopy, the effect of oxygen vacancies on domain wall conductivity can be understood comprehensively [ACS Applied Electronic Materials 3 (10): 4498-4508 (2021); ACS NANO 15 (8): 13380-13388 (2021)].

Round 2
Reviewer 2 Report
Comments and Suggestions for Authors
The authors revised the manuscript, and it can in my opinion be accepted for publication.